# Dancing to Music

**Hsin-Ying Lee**[1]    **Xiaodong Yang**[2]    **Ming-Yu Liu**[2]    **Ting-Chun Wang**[2]
**Yu-Ding Lu**[1]    **Ming-Hsuan Yang**[1]    **Jan Kautz**[2]
[1]University of California, Merced    [2]NVIDIA

## Abstract

Dancing to music is an instinctive move by humans. Learning to model the music-to-dance generation process is, however, a challenging problem. It requires significant efforts to measure the correlation between music and dance as one needs to simultaneously consider multiple aspects, such as style and beat of both music and dance. Additionally, dance is inherently multimodal and various following movements of a pose at any moment are equally likely. In this paper, we propose a synthesis-by-analysis learning framework to generate dance from music. In the analysis phase, we decompose a dance into a series of basic dance units, through which the model learns how to move. In the synthesis phase, the model learns how to compose a dance by organizing multiple basic dancing movements seamlessly according to the input music. Experimental qualitative and quantitative results demonstrate that the proposed method can synthesize realistic, diverse, style-consistent, and beat-matching dances from music.

## 1   Introduction

Does this sound familiar? Upon hearing certain genres of music, you cannot help but clap your hands, tap your feet, or swing you hip accordingly. Indeed, music inspires dances in daily life. Via spontaneous and elementary movements, people compose body movements into dances [24, 31]. However, it is only through proper training and constant practice, professional choreographers learn to compose the dance moves in a way that is both artistically elegant and rhythmic. Therefore, dance to music is a creative process that is both innate and acquired. In this paper, we propose a computational model for the music-to-dance creation process. Inspired by the above observations, we use prior knowledge to design the music-to-dance framework and train it with a large amount of paired music and dance data. This is a challenging but interesting generative task with the potential to assist and expand content creations in arts and sports, such as theatrical performance, rhythmic gymnastics, and figure skating. Furthermore, modeling how we human beings match our body movements to music can lead to better understanding of cross-modal synthesis.

Existing methods [13, 22, 26] convert the task into a similarity-based retrieval problem, which shows limited creativity. In contrast, we formulate the task from the generative perspective. Learning to synthesize dances from music is a highly challenging generative problem for several reasons. First, to synchronize dance and music, the generated dance movements, beyond realism, need to be aligned well with the given musical style and beats. Second, dance is inherently multimodal, i.e., a dancing pose at any moment can be followed by various possible movements. Third, the long-term spatio-temporal structures of body movements in dancing result in high kinematic complexity.

In this paper, we propose to synthesize dance from music through a decomposition-to-composition framework. It first learns how to move (i.e., produce basic movements) in the decomposition/analysis phase, and then how to compose (i.e., organize basic movements into a sequence) in the composition/synthesis phase. In the top-down decomposition phase, analogous to audio beat tracking of music [11], we develop a kinematic beat detector to extract movement beats from a dancing sequence. We then leverage the extracted movement beats to temporally normalize each dancing sequence

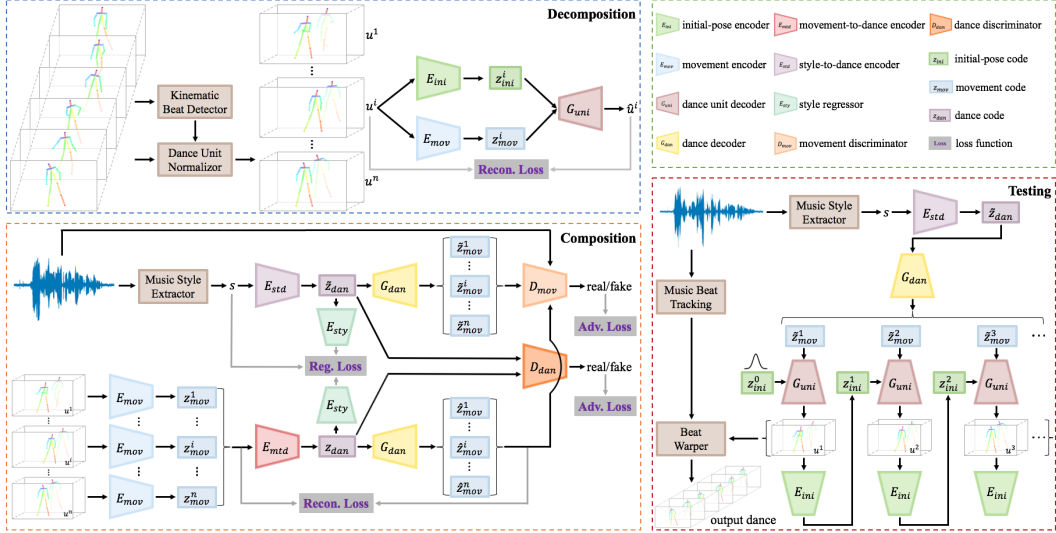

Figure 1: **A schematic overview of the decomposition-to-composition framework.** In the top-down decomposition phase (Section 3.1), we normalize the dance units that are segmented from a real dancing sequence using a kinematic beat detector. We then train the `DU-VAE` to model the dance units. In the bottom-up composition phase (Section 3.2), given a pair of music and dance, we leverage the `MM-GAN` to learn how to organize the dance units conditioned on the given music. In the testing phase (Section 3.3), we extract style and beats from the input music, then synthesize a sequence of dance units in a recurrent manner, and in the end, apply the beat warper to the generated dance unit sequence to render the output dance.

into a series of dance units. Each dance unit is further disentangled into an initial pose space and a movement space by the proposed dance unit VAE (`DU-VAE`). In the bottom-up composition phase, we propose a music-to-movement GAN (`MM-GAN`) to generate a sequence of movements conditioned on the input music. At run time given an input music clip, we first extract the style and beat information, then sequentially generate a series of dance units based on the music style, and finally warp the dance units by the extracted audio beats, as illustrated in Figure 1.

To facilitate this cross-modal audio-to-visual generation task, we collect over 360K video clips totaling 71 hours. There are three representative dancing categories in the data: "Ballet", "Zumba" and "Hip-Hop". For performance evaluation, we compare with strong baselines using various metrics to analyze realism, diversity, style consistency, and beat matching. In addition to the raw pose representation, we also visualize our results with the vid2vid model [41] to translate the synthesized pose sequences to photo-realistic videos. See our supplementary material for more details.

Our contributions of this work are summarized as follows. First, we introduce a new cross-modality generative task from music to dance. Second, we propose a novel decomposition-to-composition framework to dismantle and assemble between complex dances and basic movements conditioned on music. Third, our model renders realistic and diverse dances that match well to musical styles and beats. Finally, we provide a large-scale paired music and dance dataset, which is available along with the source code and models at our website.

## 2 Related Work

**Cross-Modality Generation.** This task explores the association among different sensory modes and leads to better understanding of human perception [17, 18, 21, 28, 30, 38, 44]. Generations between texts and images have been extensively studied, including image captioning [17, 38] and text-to-image synthesis [30, 44]. On the contrary, audio data is much less structured and thus more difficult to model its correlation with visual data. Several approaches have been developed to map vision to audio by taking visual cues to provide sound effects to videos or predict what sounds target objects can produce [8, 28, 46]. However, the generation problem from audio to visual is much less explored. Several methods focus on speech lip synchronization to predict movements of mouth landmarks from audio [18, 35]. Recent work employs LSTM based autoencoders to learn the

music-to-dance mapping [36], and uses LSTM to animate the instrument-playing avatars given an audio input of violin or piano [33].

**Audio and Vision.** The recent years have seen growing interests in cross-modal learning between audio and vision. Although hearing and sight are two distinct sensory systems, the information perceived from the two modalities is highly correlated. The correspondence between audio and vision serves as natural supervisory signals for self-supervised learning, which aims to learn feature representations by solving surrogate tasks defined from the structure of raw data [2, 4, 10, 20, 29]. Aside from representation learning, audio and visual information can be jointly used to localize the sound sources in images [3, 15, 32], predict spatial-audio from videos [23], and separate different audio-visual sources [12, 14, 27]. In addition, an audio-visual synchronization model is developed in [7] by utilizing the visual rhythm with its musical counterpart to manipulate videos.

**Human Motion Modeling.** It is challenging to model human motion dynamics due to the stochastic nature and spatio-temporal complexity. A large family of the existing work [6, 40, 42, 43] formulates motion dynamics as a sequence of 2D or 3D body keypoints, thanks to the success of human pose estimation [5]. Most of these approaches use recurrent neural networks to generate a motion sequence from a static image or a short video snippet. Some other methods consider this problem as a video generation task. Early work applies mean square loss [34] or perceptual loss [25] on raw image sequences for training. Recent methods disentangle motion and content [9, 37, 39] to alleviate the issues with holistic video generation. Another active research line is motion retargeting, which performs motion transfer between source and target subjects [1].

## 3 Music-to-Dance Generation

Our goal is to generate a sequence of dancing poses conditioned on the input music. As illustrated in Figure 1, the training process is realized by the decomposition-to-composition framework. In the top-down decomposition phase, we aim to learn how to perform basic dancing movements. For this purpose, we define and extract dance units, and introduce `DU-VAE` for encoding and decoding dance units. In the bottom-up composition phase, we target learning how to compose multiple basic movements to a dance, which conveys high-level motion semantics according to different music. So we propose `MM-GAN` for music conditioned dancing movement generation. Finally, in the testing phase, we use the components of `DU-VAE` and `MM-GAN` to recurrently synthesize a long-term dance in accordance with the given music.

### 3.1 Learning How to Move

In the music theory, beat tracking is usually derived from **onset** [11], which can be defined as the start of a music note, or more formally, the beginning of an acoustic event. Current audio beat detection algorithms are mostly based on detecting onset using a spectrogram $S$ to capture the frequency domain information. We can measure the change in different frequencies by $S_{\text{diff}}(t, k) = |S(t, k)| - |S(t - 1, k)|$, where $t$ and $k$ indicate the time step and quantized frequency, respectively. More details on music beat tracking can be found in [11]. Unlike music, the kinematic beat of human movement is not well defined. We usually perceive the sudden motion deceleration or **offset** as a kinematic beat. A similar observation is also recently noted in [7].

We develop a kinematic beat detector to detect when a movement drastically slows down. In practice, we compute the motion magnitude and angle of each keypoint between neighboring poses, and track the magnitude and angle trajectories to spot when a dramatic decrease in the motion magnitude or a substantial change in the motion angle happens. Analogous to the spectrogram $S$, we can construct a matrix $D$ to capture the motion changes in different angles. For a pose $p$ of frame $t$, the difference in a motion angle bin $\theta$ is summed over all joints:

$$D(t, \theta) = \sum_i |p_t^i - p_{t-1}^i| Q(p_t^i, p_{t-1}^i, \theta), \tag{1}$$

where $Q$ is an indicator function to quantize the motion angles. Then, the changes in different motion angles can be computed by:

$$D_{\text{diff}}(t, \theta) = |D(t, \theta)| - |D(t - 1, \theta)|. \tag{2}$$

This measurement captures abrupt magnitude decrease in the same direction, as well as dramatic change of motion direction. Finally, the kinematic beats can be detected by thresholding $D_{\text{diff}}$.

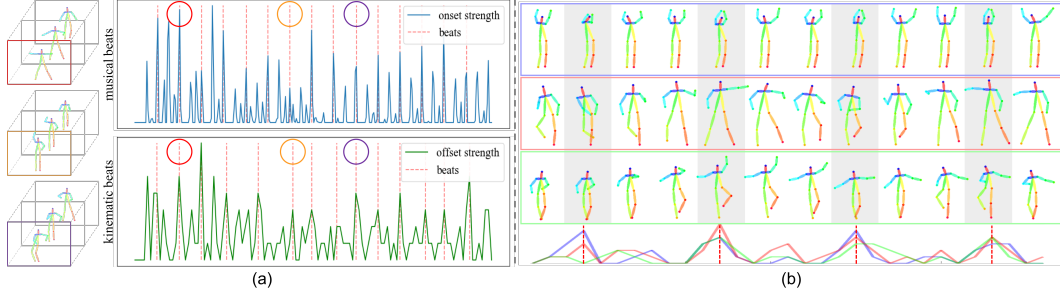

Figure 2: **(a) Extraction of beats from music and dance.** For music, periodical beats are extracted by the onset strength. For dance, we compute the offset strength and extract kinematic beats. We illustrate three example frames corresponding to the aligned music and kinematic beats: lateral arm raising (red), hand raising (yellow), and elbow pushing out (purple). **(b) Examples of dance units.** Every dance unit is of the same length and with kinematic beats assigned in the specific beat times.

However, in reality, people do not dance to every musical beat. Namely, each kinematic beat needs to align with a musical beat, yet it is unnecessary to fit every musical beat while dancing. Figure 2(a) shows the correspondence between the extracted musical beats by a standard audio beat tracking algorithm [11] and the kinematic beats by our kinematic beat detector. Most of our detected kinematic beats match the musical beats accurately.

Leveraging the extracted kinematic beats, we define the dance unit in this work. As illustrated in Figure 2(b), a dance unit is a temporally standardized short snippet, consisting of a fixed number of poses, whose kinematic beats are normalized to several specified beat times with a constant beat interval. A dance unit captures basic motion patterns and serves as atomic movements, which can be used to constitute a complete dancing sequence. Another benefit of introducing the dance unit is that, with temporal normalization of beats, we can alleviate the beat factor and simplify the generation to focus on musical style. In the testing phase, we incorporate the music beats to warp or stretch the synthesized sequence of dance units.

After normalizing a dance into a series of dance units, the model learns how to perform basic movements. As shown in the decomposition phase of Figure 1, we propose to disentangle a dance unit into two latent spaces: an initial pose space $\mathcal{Z}_{ini}$ capturing the single initial pose, and a movement space $\mathcal{Z}_{mov}$ encoding the motion that is agnostic of the initial pose. This disentanglement is designed to facilitate the long-term sequential generation, i.e., the last pose of a current dance unit can be used as the initial pose of the next one, so that we can continuously synthesize a full long-term dance. We adopt the proposed DU-VAE to perform the disentangling. It consists of an initial-pose encoder $E_{ini}$, a movement encoder $E_{mov}$, and a dance unit decoder $G_{uni}$. Given a dance unit $u \in \mathcal{U}$, we exploit $E_{ini}$ and $E_{mov}$ to encode it into the two latent codes $z_{ini} \in \mathcal{Z}_{ini}$ and $z_{mov} \in \mathcal{Z}_{mov}$: $\{z_{ini}, z_{mov}\} = \{E_{ini}(u), E_{mov}(u)\}$. As $G_{uni}$ should be able to reconstruct the two latent codes back to $\hat{u}$, we enforce a reconstruction loss on $u$ and a KL loss on the initial pose space and movement space to enable the reconstruction after encoding and decoding:

$$
\begin{aligned}
L_{\mathrm{recon}}^{u} &= \mathbb{E}[\|G_{uni}(z_{ini}, z_{mov}) - u\|_1], \\
L_{\mathrm{KL}}^{u} &= \mathbb{E}[\mathrm{KL}(\mathcal{Z}_{ini}\|N(0,\mathrm{I}))] + \mathbb{E}[\mathrm{KL}(\mathcal{Z}_{mov}\|N(0,\mathrm{I}))],
\end{aligned}
\tag{3}
$$

where $\mathrm{KL}(p\|q) = -\int p(z) \log \frac{p(z)}{q(z)} \mathrm{d}z$. We apply the KL loss on $\mathcal{Z}_{ini}$ for random sampling of the initial pose at test time, and the KL loss on $\mathcal{Z}_{mov}$ to stabilize the composition training in the next section. With the intention to encourage $E_{mov}$ to disregard the initial pose and focus on the movement only, we design a shift-reconstruction loss:

$$
L_{\mathrm{recon}}^{\mathrm{shift}} = \mathbb{E}[\|G_{uni}(z_{ini}, E_{mov}(u')) - u\|_1],
\tag{4}
$$

where $u'$ is a spatially shifted $u$. Overall, we jointly train the two encoders $E_{ini}$, $E_{mov}$, and one decoder $G_{uni}$ of DU-VAE to optimize the total objective in the decomposition:

$$
L_{\mathrm{decomp}} = L_{\mathrm{recon}}^{u} + \lambda_{\mathrm{KL}}^{u} L_{\mathrm{KL}}^{u} + \lambda_{\mathrm{recon}}^{\mathrm{shift}} L_{\mathrm{recon}}^{\mathrm{shift}},
\tag{5}
$$

where $\lambda_{\mathrm{KL}}^{u}$ and $\lambda_{\mathrm{recon}}^{\mathrm{shift}}$ are the weights to control the importance of KL and shift-reconstruction terms.

## 3.2 Learning How to Compose

Since a dance consists of a sequence of movement units in a particular arrangement, different combinations can represent different expressive semantics. Based on the movement space $\mathcal{Z}_{mov}$ disentangled from the aforementioned decomposition, the composition model learns how to meaningfully compose a sequence of basic movements into a dance conditioned on the input music.

As demonstrated in the composition phase of Figure 1, the proposed MM-GAN is utilized to bridge the semantic gap between low-level movements and high-level music semantics. Given a dance, we first normalize it into a sequence of $n$ dance units $\{u^i\}_{i=1}^n$, and then encode them to the latent movement codes $\{z_{mov}^i\}_{i=1}^n$, as described in the decomposition phase. In this context, $\{\cdot\}$ denotes a temporally ordered sequence, for notational simplicity, we skip the temporal number $n$ in the following. We encode $\{z_{mov}^i\}$ to a dancing space $\mathcal{Z}_{dan}$ with a movement-to-dance encoder $E_{mtd}$: $\{z_{mov}^i\} \rightarrow z_{dan}$, and reconstruct $z_{dan}$ back to $\{\hat{z}_{mov}^i\}$ with a recurrent dance decoder $G_{dan}$. For the corresponding music, we employ a music style extractor to extract the style feature $s$ from the audio feature $a$. Since there exists no robust style feature extractor given our particular needs, we train a music style classifier on the collected music for this task. We encode $s$ along with a noise vector $\epsilon$ to a latent dance code $\tilde{z}_{dan} \in \mathcal{Z}_{dan}$ using a style-to-dance encoder $E_{std}$: $(s, \epsilon) \rightarrow \tilde{z}_{dan}$, and then make use of $G_{dan}$ to decode $\tilde{z}_{dan}$ to a latent movement sequence $\{\tilde{z}_{mov}^i\}$.

It is of great importance to ensure the alignments among movement distributions and among dance distributions that are respectively produced by real dance and corresponding music. To this end, we use adversarial training to match the distributions between $\{\hat{z}_{mov}^i\}$ encoded and reconstructed from the real dance units and $\{\tilde{z}_{mov}^i\}$ generated from the associated music. As the audio feature $a$ contains low-level musical properties, we make the decision conditioned on $a$ to further encourage the correspondence between music and dance:

$$L_{\text{adv}}^m = \mathbb{E}[\log D_{mov}(\{\hat{z}_{mov}^i\}, a) + \log(1 - D_{mov}(\{\tilde{z}_{mov}^i\}, a))], \qquad (6)$$

where $D_{mov}$ is the discriminator that tries to distinguish between the movement sequences that are generated from real dance and music. Compared to the distribution of raw data, such as poses, it is more difficult to model the distribution of latent code sequences, or, $\{z_{mov}^i\}$ in our case. We thus adopt an auxiliary reconstruction task on the latent movement sequences to facilitate training:

$$L_{\text{recon}}^m = \mathbb{E}[\|\{\hat{z}_{mov}^i\} - \{z_{mov}^i\}\|_1]. \qquad (7)$$

For the alignment between latent dance codes, we apply a discriminator $D_{dan}$ to differentiate the dance codes encoded from real dance and music, and enforce a KL loss on the latent dance space:

$$
\begin{aligned}
L_{\text{adv}}^d &= \mathbb{E}[\log D_{dan}(z_{dan}) + \log(1 - D_{dan}(\tilde{z}_{dan}))], \\
L_{\text{KL}}^d &= \mathbb{E}[\text{KL}(\mathcal{Z}_{dan}\|N(0, \text{I}))].
\end{aligned}
\qquad (8)
$$

As the style feature $s$ embodies high-level musical semantics that should be reflected in the dance code $z_{dan}$, we therefore use a style regressor $E_{sty}$ on the latent dance codes to reconstruct $s$ to further encourage the alignment between the styles of music and dance:

$$L_{\text{recon}}^s = \mathbb{E}[\|E_{sty}(z_{dan}) - s\|_1 + \|E_{sty}(\tilde{z}_{dan}) - s\|_1]. \qquad (9)$$

Overall, we jointly train the three encoders $E_{mtd}$, $E_{std}$, $E_{sty}$, one decoder $G_{dan}$, and two discriminators $D_{mov}$, $D_{dan}$ of MM-GAN to optimize the full objective in the composition:

$$L_{\text{comp}} = L_{\text{recon}}^m + \lambda_{\text{recon}}^s L_{\text{recon}}^s + \lambda_{\text{adv}}^m L_{\text{adv}}^m + \lambda_{\text{adv}}^d L_{\text{adv}}^d + \lambda_{\text{KL}}^d L_{\text{KL}}^d, \qquad (10)$$

where $\lambda_{\text{recon}}^s$, $\lambda_{\text{adv}}^m$, $\lambda_{\text{adv}}^d$, and $\lambda_{\text{KL}}^d$ are the weights to control the importance of related loss terms.

## 3.3 Testing Phase

As shown in the testing phase of Figure 1, the final network at run time consists of $E_{ini}$, $G_{uni}$ learned in the decomposition and $E_{sty}$, $G_{dan}$ trained in the composition. Given a music clip, we first track the beats and extract the style feature $s$. We encode $s$ with a noise $\epsilon$ into a latent dance code $\tilde{z}_{dan}$ by $E_{std}$ and then decode $\tilde{z}_{dan}$ to a movement sequence $\{\tilde{z}_{mov}^i\}$ by $G_{dan}$. To compose a complete dance, we randomly sample an initial pose code $z_{ini}^0$ from the prior distribution, and then recurrently generate a full sequence of dance units using $z_{ini}^0$ and $\{\tilde{z}_{mov}^i\}$. The initial pose code $z_{ini}^i$ of the next dance unit can be encoded from the last frame of the current dance unit:

$$u^i = G_{uni}(z_{ini}^{i-1}, z_{mov}^i), \qquad z_{ini}^i = E_{ini}(u^i(-1)), \qquad (11)$$

where $u^i(-1)$ is the last frame of the $i$th dance unit. With these steps, we can continuously and seamlessly generate a long-term dancing sequence fitting into the input music. Since the beat times are normalized in each dance unit, we in the end warp the generated sequence of dance units by aligning their kinematic beats with the extracted music beats to produce the final full dance.

## 4    Experimental Results

We conduct extensive experiments to evaluate the proposed decomposition-to-composition framework. We qualitatively and quantitatively compare our method with several baselines on various metrics including motion realism, style consistency, diversity, multimodality, and beat coverage and hit rate. Experimental results reveal that our method can produce more realistic, diverse, and music-synchronized dances. More comparisons are provided in the supplementary material. Note that we could not include music in the embedded animations of this PDF, but the complete results with music can be found in the supplementary video.

### 4.1    Data Collection and Processing

Since there exists no large-scale music-dance dataset, we collect videos of three representative dancing categories from the Internet with the keywords: "Ballet", "Zumba", and "Hip-Hop". We prune the videos with low quality and few motion, and extract clips in 5 to 10 seconds with full pose estimation results. In the end, we acquire around 68K clips for "Ballet", 220K clips for "Zumba", and 73K clips for "Hip-Hop". The total length of all the clips is approximately 71 hours. We extract frames with 15 fps and audios with 22 kHz. We randomly select 300 music clips for testing and the rest used for training.

**Pose Processing.** OpenPose [5] is applied to extract 2D body keypoints. We observe that in practice some keypoints are difficult to be consistently extracted in the wild web videos and some are less related to dancing movements. So we finally choose 14 most relevant keypoints to represent the dancing poses, i.e., nose, neck, left and right shoulders, elbows, wrists, hips, knees, and ankles. We interpolate the missing detected keypoints from the neighboring frames so that there are no missing keypoints in all extracted clips.

**Audio Processing.** We use the standard MFCC as the music feature representation. The audio volume is normalized using root mean square with FFMPEG. We then extract the 13-dimensional MFCC feature, and concatenate it with its first temporal derivatives and log mean energy of volume into the final 28-dimensional audio feature.

### 4.2    Implementation Details

Our model is implemented in PyTorch. We use the gated recurrent unit (GRU) to build encoders $E_{mov}$, $E_{mtd}$ and decoders $G_{uni}$, $G_{dan}$. Each of them is a single-layer GRU with 1024 hidden units. $E_{ini}$, $E_{std}$, and $E_{sty}$ are encoders consisting of 3 fully-connected layers. $D_{dan}$ and $D_{mov}$ are discriminators containing 5 fully-connected layers with layer normalization. We set the latent code dimensions to $z_{ini} \in R^{10}$, $z_{mov} \in R^{512}$, and $z_{dan} \in R^{512}$. In the decomposition phase, we set the length of a dance unit as 32 frames and the number of beat times within a dance unit as 4. In the composition phase, each input sequence contains 3 to 5 dance units. For training, we use the Adam optimizer [19] with batch size of 512, learning rate of 0.0001, and exponential decay rates $(\beta_1, \beta_2) = (0.5, 0.999)$. In all experiments, we set the hyper-parameters as follows: $\lambda_{\mathrm{KL}}^u = \lambda_{\mathrm{KL}}^d = 0.01$, $\lambda_{\mathrm{recon}}^{\mathrm{shift}} = 1$, $\lambda_{\mathrm{adv}}^d = \lambda_{\mathrm{adv}}^m = 0.1$, and $\lambda_{\mathrm{recon}}^s = 1$. Our data, code and models are publicly available at our website.

### 4.3    Baselines

Generating dance from music is a relatively new task from the generative perspective and thus few methods have been developed. In the following, we compare the proposed algorithm to the several strong baseline methods. As our comparisons mainly target generative models, we present the results of traditional retrieval-based method in the supplementary material.

**LSTM.** We use LSTM as our deterministic baseline. Similar to the recent work on mapping audio to arm and hand dynamics [33], the model takes audio features as inputs and produces pose sequences.

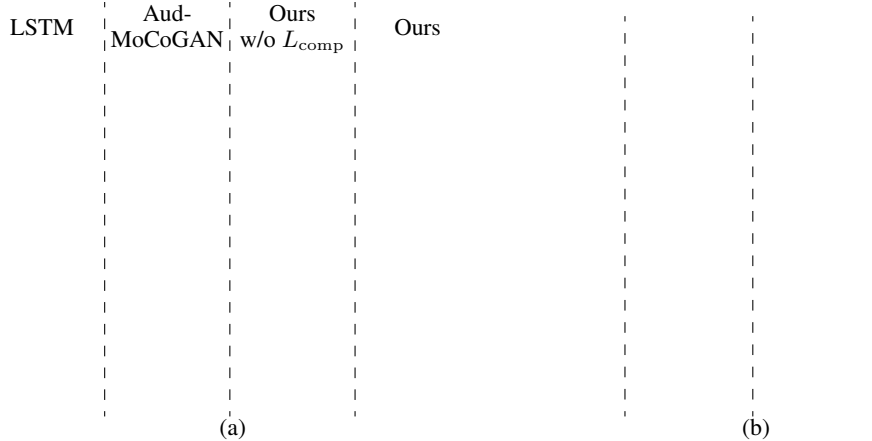

Figure 3: **(a) Comparison of the generated dances.** LSTM produces dances that tend to collapse to static poses. Aud-MoCoGAN generates jerking dances that are prone to repeating the same movements. Ours w/o $L_{\text{comp}}$ produces realistic movements, yet the combinations of movements are often unnatural. Compared to the baselines, our results are realistic and coherent. **(b) Examples of multimodal generation.** Dances in each column are generated by our method using the same music clip and initial pose. This figure is best viewed via Acrobat Reader. Click each column to play.

**Aud-MoCoGAN.** MoCoGAN [37] is a video generation model, which maps a sequence of random vectors containing the factors of fixed content and stochastic motion to a sequence of video frames. We modify this model to take extracted audio features on style and beat as inputs in addition to noise vectors. To improve the quality, we use multi-scale discriminators and apply curriculum learning to gradually increase the dance sequence length.

**Ours w/o $L_{\text{comp}}$.** This model ablates the composition phase and relies on the decomposition phase. In addition to the original `DU-VAE` for decomposition, we enforce the paired music and dance unit to stay close when mapped in the latent movement space. At test time, we map a music clip into the movement space, and then recurrently generate a sequence of dance units by using the last pose of one dance unit as the first pose of the next one.

## 4.4 Qualitative Comparisons

We first compare the quality of synthesized dances by different methods. Figure 3(a) shows the dances generated from different input music. We observe that the dances generated by LSTM tend to collapse to certain poses regardless of the input music or initial pose. The deterministic nature of LSTM hinders it from learning the desired mapping to the highly unconstrained dancing movements. For Aud-MoCoGAN, the generated dances contain apparent artifacts such as twitching or jerking in an unnatural way. Furthermore, the synthesized dances tend to be repetitive, i.e., performing the same movement throughout a whole sequence. This may be explained by the fact that Aud-MoCoGAN takes all audio information including style and beat as input, of which correlation with dancing movements is difficult to learn via a single model. Ours w/o $L_{\text{comp}}$ can generate smoother dances compared to the above two methods. However, since the dance is simply formed by a series of independent dance units, it is easy to observe incoherent movements. For instance, the third column in Figure 3(a) demonstrates the incoherent examples, such as mixing dance with different styles (top), an abrupt transition between movements (middle), and unnatural combination of movements (bottom). In contrast, the dances generated by our full model are more realistic and coherent. As demonstrated in the fourth column in Figure 3(a), the synthesized dances consist of smooth movements (top), consecutive similar movements (middle), and a natural constitution of raising the left hand, raising the right hand, and raising both hands (bottom).

We also analyze two other important properties for the music-to-dance generation: multimodality and beat matching. For multimodality, our approach is able to generate diverse dances given the same music. As shown in Figure 3(b), each column shows various dances that are synthesized from the same music and the same initial pose. For beat matching, we compare the kinematic beats extracted from the generated dances and their corresponding input music beats. Most kinematic beats of our generated dances occur at musical beat times. Figure 4 visualizes two short dancing snippets which

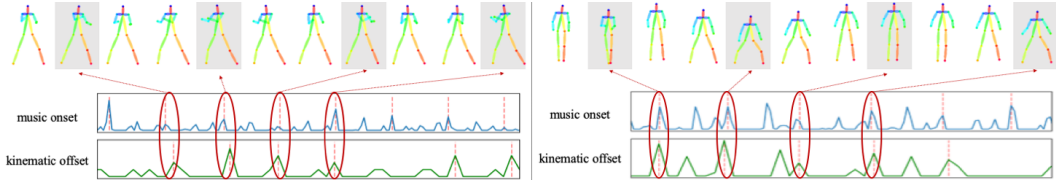

Figure 4: **Examples of beat matching between music and generated dances.** We show two generated dances with the extracted kinematic offsets as well as music onsets from input music. The red dashes on the onset and offset graphs indicate the extracted musical beats and kinematic beats. The consecutive matched beats correspond to clapping hands on left and right alternatively (left), and squatting down repetitively (right).

align with their musical beats, including clapping hands to left and right alternatively, and squatting down repetitively. More demonstrations with music, such as long-term generation, mixing styles and photo-realistic translation, are available in the supplementary video.

### 4.5 Quantitative Comparisons

**Motion Realism and Style Consistency.** Here we perform a quantitative evaluation of the realism of generated movements and the style consistency of synthesized dances to the input music. We conduct a user study using a pairwise comparison scheme. Specifically, we evaluate generated dances from the four methods as well as real dances on 60 randomly selected testing music clips. Given a pair of dances with the same music clip, each user is asked to answer two questions: "Which dance is more realistic regardless of music?" and "Which dance matches the music better?". We ask each user to compare 20 pairs and collect results from a total of 50 subjects.

Figure 5 shows the user study results, where our approach outperforms the baselines on both motion realism and style consistency. It is consistently found that LSTM and Aud-MoCoGAN generate dances with obvious artifacts and result in low preferences. Although ours w/o $L_{comp}$ can produce high-quality dance units, the simple concatenation of independent dance units usually makes the synthesized dance look unnatural. This is also reflected in the user study, where $61.2\%$ prefer the full solution in term of motion realism, and $68.3\%$ in style consistency. Compared to the real dances, $35.7\%$ of users prefer our approach in term of motion realism and $28.6\%$ in style consistency. Note that the upper bound is $50.0\%$ when comparing to the real dances. The performance of our method can be further improved with more training data.

In addition to the subjective test, we evaluate the visual quality following Fréchet Inception Distance (FID) [16] by measuring how close the distribution of generated dances is to the real. As there exists no standard feature extractor for pose sequences, we train an action classifier on the collected data of three categories as the feature extractor. Table 1 shows the average results of 10 trials. Overall, the FID of our generated dances is much closer to the real ones than the other evaluated methods.

**Beat Coverage and Hit Rate.** In addition to realism and consistency, we evaluate how well the kinematic beats of generated dances match the input music beats. Given all input music and generated dances, we gather the number of total musical beats $B_m$, the number of total kinematic beats $B_k$, and the number of kinematic beats that are aligned with musical beats $B_a$. We use two metrics for evaluation: (i) beat coverage $B_k/B_m$ measures the ratio of kinematic beats to musical beats, (ii) beat hit rate $B_a/B_k$ is the ratio of aligned kinematic beats to total kinematic beats.

As shown in Table 1, our approach generates very similar beat coverage as real dances, indicating our synthesized dances can naturally align with the musical rhythm. Note that for beat coverage, it is not the higher the better, but depends on the different dancing styles. Ours w/o $L_{comp}$ has a higher beat hit rate than our full model as the latter takes coherence between movements into account, which may sacrifice beat hit rate of individual movements. There are two main reasons for the relatively low beat hit rate of real dances. First, the data is noisy due to automatic collection process and imperfect pose extraction. Second, our kinematic beat detector is an approximation, which may not be able to capture all subtle motions that can be viewed as beat points by human beings.

**Diversity and Multimodality.** We evaluate the diversity among dances generated by various music and the multimodality among dances generated from the same music. We use the average feature distance similar to [45] as the measurement. In addition, we use the same feature extractor as used

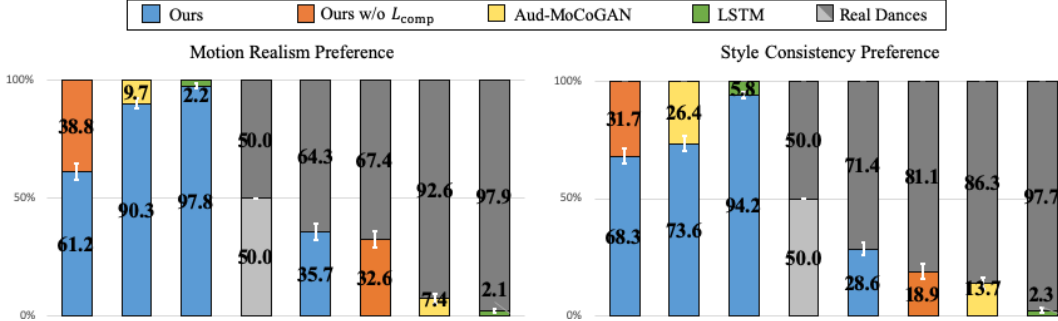

Figure 5: **Preference results on motion realism and style consistency.** We conduct a user study to ask subjects to select the dances that *are more realistic regardless of music* and *better match the style of music* through pairwise comparisons. Each number denotes the percentage of preference on the corresponding comparison pair.

| Method | FID | Beat Coverage | Beat Hit Rate | Diversity | Multimodality |
|---|---|---|---|---|---|
| Real Dacnes | $5.9 \pm 0.4$ | 39.3 % | 51.6 % | $53.5 \pm 1.9$ | - |
| LSTM | $73.8 \pm 4.1$ | 1.4 % | 0.8 % | $24.5 \pm 1.4$ | - |
| Aud-MoCoGAN | $21.7 \pm 0.8$ | 23.9 % | 54.8 % | $45.8 \pm 1.3$ | $27.3 \pm 1.3$ |
| Ours w/o $L_{\mathrm{comp}}$ | $14.8 \pm 1.1$ | 37.8 % | **72.4 %** | $49.7 \pm 2.0$ | $\mathbf{51.4 \pm 0.8}$ |
| Ours | $\mathbf{12.8 \pm 0.8}$ | **39.4 %** | 65.1 % | $\mathbf{53.2 \pm 2.5}$ | $47.8 \pm 0.9$ |

Table 1: **Comparison of realism.** FID evaluates the visual quality by measuring the distance between the distributions of real and synthesized dances. **Comparison of beat coverage and hit rate.** We quantify the correspondence between input music beats and generated kinematic beats. Beat coverage measures the ratio of total kinematic beats to total musical beats. Beat hit rate measures the ratio of kinematic beats that are aligned with musical beats to total kinematic beats. **Comparison of diversity and multimodality.** We evaluate the diversity and multimodality using average feature distances. We use diversity to refer to the variations among a set of dances, while multimodality to reflect the variations of generated dances given the same input music.

in measuring FID. For diversity, we generate 50 dances from different music on each trial, then compute the average feature distance between 200 random combinations of them. For multimodality, it compares the ability to generate diverse dances conditioned on the same music. We measure the average distance between all combinations of 5 dances generated from the same music.

Table 1 shows the average results of 10 trials for diversity and 500 trials for multimodality. The multimodality score of LSTM is not reported since LSTM is a deterministic model and incapable of multimodal generation. Our generated dances achieve comparable diversity score to real dances and outperform Aud-MoCoGAN on both diversity and multimodality scores. Ours w/o $L_{\mathrm{comp}}$ obtains a higher score on multimodality since it disregards the correlation between consecutive movements and is free to combine them with the hurt to motion realism and style consistency. However, the proposed full model performs better in diversity, suggesting that the composition phase in training enforces movement coherence at no cost of diversity.

## 5    Conclusions

In this work, we have proposed to synthesize dances from music through a decomposition-to-composition learning framework. In the top-down decomposition phase, we teach the model how to generate and disentangle the elementary dance units. In the bottom-up composition phase, we direct the model to meaningfully compose the basic dancing movements conditioned on the input music. We make use of the kinematic and musical beats to temporally align generated dances with accompanying music. Extensive qualitative and quantitative evaluations demonstrate that the synthesized dances by the proposed method are not only realistic and diverse, but also style-consistent and beat-matching. In the future work, we will continue to collect and incorporate more dancing styles, such as pop dance and partner dance.

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
