[Supplementary Material]

# Dance to Music
## SUPPLEMENTARY MATERIAL

**Hsin-Ying Lee**[1]    **Xiaodong Yang**[2]    **Ming-Yu Liu**[2]    **Ting-Chun Wang**[2]
**Yu-Ding Lu**[1]    **Ming-Hsuan Yang**[1]    **Jan Kautz**[2]
[1]University of California, Merced    [2]NVIDIA

In this supplementary material, we show the comparison with traditional retrieval-based method for music to dance generation. We also demonstrate a video with music to provide a better visual and auditory experience of the synthesized dances.

## A    Comparison to Retrieval Methods

In addition to creativity, we present some additional limitations of the traditional similarity-based retrieval methods. We adopt the proposed method in (1) with the same 28-dim audio features as used in our framework. We also follow (1) to apply spline interpolation to smooth the junction between consecutive retrieved segments. Figure 6 demonstrates some representative examples from the retrieval. Since music within a clip tends to have similar audio features, the retrieved segments are likely to be the same, as shown in the first three columns. If adding a restriction on choosing the same segment, the transition between different segments can be physically unnatural, as shown in the fourth column. Moreover, the simple concatenation of different segments without global modeling may result in inconsistent segments, as shown in the last two columns. Finally, the retrieval-based methods cannot guarantee to match the input music beats. In term of beat coverage, the retrieval-based method gets 36.7%, while the proposed method scores 39.4%.

Figure 6: **Representative examples of retrieval methods.** First three examples show that adjacent music is likely to retrieve repetitive segments. Last three examples present the inconsistency when selecting different segments. The different boundary colors indicate consecutive segments in the final dance. This figure is best viewed via Acrobat Reader. Click each column to play.

## References

[1] M. Lee, K. Lee, and J. Park. Music similarity-based approach to generating dance motion sequence. *Multimedia Tools and Applications*, 2013. 1