[Reviews · NeurIPS 2019]

Reviewer 1



a. I like the idea of using two VAEs to model dance at different levels. Generating complex sequence is very challenging. So decomposing generative process into stages makes a lot of sense. b. The proposed dance synthesis model uses autoregressive approach to generate dance sequence, simplifying the sequence generation process. The low level VAE decomposes dance unit into initial pose and movement. The high level VAE models movement sequence, and shares a latent space with the output of music VAE. c. The adversarial losses and reconstruction losses are carefully design to improve the naturalness of generated dance. d. The video demo clearly shows that the proposed model outperforms the baselines. e. Paper organization and presentation are good.

Reviewer 2



Originality - The work is more than moderately original. Quality - The quality of the work/experiment/evaluation is high. Clarity - The paper is structured well and written nicely. But I have several comments as below. Significance - The work is moderately significant. The impact on the same task would be big but is limited to the area around it. ---- comments ---- Title - I would like to strongly suggest to change the title. "Dance to music" can be a nickname of this paper but not a title. I don't think I need to list everything about a good title. Abstract - the "top-down" and "bottom-up" doesn't add any information and therefore seem unnecessary. I can't think of any non-top-down analysis and I was actually even confused by these words because I thought it may mean some very special kind of analysis or synthesis. L21 - "Inspired by the above observations" -- which observations exactly? It seems unclear to me. L31 - Overall in this paper, "multimodality" is undefined and simply replaced with "diversity" because that's what it really means. In the experiment, there are two different kinds of diversity measures (and only by then I was sure that it means diversity), but they can be called as "XX diversity" and "YY diversity". Multimodality as a mean of diversity is commonly used in GAN literature, but they are more likely to mean something else (e.g., multi-domain like audio and video), therefore it is confusing. L67 and L77 - those two concepts are not in parallel. Also, overall, the two paragraphs seem somehow redundant and may be compressed if the authors need more space. L107 - a fixed number of poses - how many? Overall in Section 3.1 and 3.2 - a clearer and more explicit hypothesis and assumption(s) would be nice. By building up this structure and planning the proposed approach, what is assumed? Like probably all the other works, there are some assumptions that allow the authors to model the whole system in this way, e.g., using VAEs for them, some hyper parameters, etc. It is actually already good, but I think it can be slightly improved. L146 - More detail on the music style classifier is necessary. Or at least a reference. I was surprised by not finding this in the supplementary material. L192 - L198 - Looks like a legit choice, but again, the details of these systems are absolutely necessary. L205 - L221 - Although it's not bad to have this information, at the end of the day, these are completely subjective and one can write the exact same contents with cherry-picked examples. I think this should be more compact and probably mentioned only after all the quantitive results are shown. L223 - Again, I don't see why we should call it multimodality and not diversity. Section 4.3 - It would be nicer if it is more explicit that this quantitative result is still from a subjective test. L237, L240 - "Style consistency" can mean a lot of things, e.g., the consistency over time. Won't there be a better way to describe it? L238 - 50 subjects - who are they? L250 - L252 - the action classifier should be elaborated much, much more than this. Reference and background - "J. Lee et al., 2018 Nov" could be discussed, too, especially considering it's timeliness.

Reviewer 3



Learning to generate dance according to a given piece of music is an interesting task, and could be benificial to artists in related areas. Both adversarial learning and reconstruction loss are widely used in various generaiton tasks, they are never applied to this new task before this work. Therefore, I recognize the innovation in terms of methodology made by this application work. Evaluation include both quantitative results and qualitative results. From the quantitative results (on automatic metrics and human judgment), it looks like the improvement over the selected baselines is significant. The authors also provide a video in supplementary material and show how the dance generated visually. Overall, I think the paper makes decent contributions to AI research and industry, however, I have several concerns (suggestions): 1. The authors hilghlight their innovation on decomposition of dance session to dance units. However, from their descriptions in the supplementary material, they just divide the dance session to small pieces with each 32 frames (2 seconds). Thus my understanding is that the dance unit is independent with kinematic beat or onset strength. Then what's special for the dance unit? 2 Dance generation is not totally new. The following work studies the same problem with deep learning techniques, but is ignored by the authors: a. Generative Choreography using Deep Learning b. Dance with Melody: An LSTM-autoencoder Approach to Music oriented Dance Synthesis I suggest the authors to compare their method with these existing ones. 3. Long sequence generation is a big challenge for DL based models due to exposure bias. It is common that the model will output similar units (e.g., poses in the context of dance generation) after a few steps. Therefore, I doubt about if the proposed method can really generate long sequences, since 20 seconds is not long. 4. Poses in the selected dance styles are relatively simple. Have you tried generation of any pop dances that with complicated poses?

[Author Response · NeurIPS 2019]

We thank the reviewers for the constructive comments and suggestions. Below we address the main issues. We will fix all other minor issues in the revised paper.

**R1, R3: Dance unit.** We will elaborate on the dance units, which are not a simple division of a dance sequence. We first extract kinematic beats and then align the extracted beat times to the pre-defined time steps (i.e., 8, 16, 24 and 32 in a dance unit of length 32). Finally, we interpolate and extrapolate the original pose sequences to obtain the temporally normalized dance units, which are further encoded into the disentangled initial pose space and movement space.

**R2: Title.** Thanks for the suggestion. We will change the title in the revision.

**R2: Term usage.** (1) **Abstract:** We agree with the comment and will remove the terms "top-down" and "bottom-up" in the abstract. (2) **Multimodality:** We defined the term "multimodality" in L272 and in the caption of Table 1. However, we agree that the usage of this term might be ambiguous. We will replace it with more explicit expressions (e.g., "overall diversity" and "instance diversity"). (3) **Style-consistency:** We will replace it with a more explicit term such as "music-dance consistency"or "music-dance matching".

**R2: Unclear observations.** The observations in L21 refer to "dance to music is a creative process that is both innate and acquired". Similarly, we leverage our prior knowledge (innate) to design the framework and then use a data-driven learning process, which corresponds to the acquired properties.

**R2: What is assumed on the model.** To facilitate the training process, we assume that the latent factors of the music-to-dance generation process can be disentangled to two components: beat and style. In the decomposition, we learn how to dance according to the beat. In the composition, we learn how to dance to the beat and style at the same time.

**R2: More details.** Our code and trained model will be released, where all implementation details can be found. (1) **Number of poses:** We use 32 frames for each dance unit in our experiments (i.e., two seconds at 16 fps). (2) **Music style classifier:** We have evaluated several methods to extract features: features from a pre-trained network SoundNet, from a music autoencoder, and from a music classifier. We adopt the music classifier as our music style extractor due to the capability to better separate different types of music. This classifier consists of multiple fully-connected layers and takes as inputs the MFCC features extracted from music of various types. The training data are the audios from videos used for training our dance model, totaling 360K clips and three categories. (3) **Baselines:** LSTM baseline is basically the same as [28] (see Figure 5 in [28]). It takes a sequence of extracted audio features as inputs and directly predicts a sequences of poses. Aud-MoCoGAN is an extention of MoCoGAN [31]. The original MoCoGAN takes a sequence of random variables as inputs. We use audio features (MFCC) and beats (a sequence of binary variables) as additional inputs. (4) **Action classifier:** We utilize an RNN to take as inputs the pose sequences of arbitrary length, and append multiple fully-connected layers on the last hidden state to perform classification. We use the features from the last fully-connected layer to compute the FID. (5) **Subjective test:** We conduct the subjective test online. The professional backgrounds (high-school students, doctors, professors) and ages (from 15 to 61) of the subjects are diverse.

**R2: Quantitative and qualitative results.** We will reorganize the section order according to the suggestion. Since the trained model will be released, it is easy to qualitatively evaluate the results. For the quantitative results, only motion realism and style consistency are from the subjective test. Other evaluation metrics including FID, beat hit-rate, beat coverage, diversity, and multimodality are all measured quantitatively.

**R2, R3: More references.** The LSTM-based methods are similar to our first baseline and suffer from diversity and multimodality. We will cite and discuss these papers in the revised manuscript.

**R3: Long sequence generation.** In our framework, we can generate up to five consecutive dance units (about 10 seconds) in each step, and a sequence of arbitrary length can be seamlessly generated in an iterative manner. In this work, we use the last pose of the previous step as the initial pose of the next step, as shown in the bottom right of Figure 1 and Eq. (9). We will design methods to better capture the long-term relationship among sub-sequences (e.g., through a hierarchical recurrent network) in our future work.

**R3: Complicated dance.** In term of complicated poses, the hip-hop dance in our selected dance styles is quite complicated. We will continue to collect and incorporate more dance styles including pop dance and partner dance.



[Meta-Review · NeurIPS 2019]

This paper proposes a novel approach to generating dance from music. The considered problem is interesting and exciting. The proposed two-phase generation approach is sensible and clever, with impressive quantitative and qualitative experimental results. This paper will be of rather wide interest to the community.